# Ochratoxin A: Overview of Prevention, Removal, and Detoxification Methods

**DOI:** 10.3390/toxins15090565

**Published:** 2023-09-08

**Authors:** Lijuan Ding, Meihua Han, Xiangtao Wang, Yifei Guo

**Affiliations:** Institute of Medicinal Plant Development, Chinese Academy of Medical Sciences & Peking Union Medical College, No. 151, Malianwa North Road, Haidian District, Beijing 100193, China; dlj990818@163.com (L.D.); hanmeihua727@163.com (M.H.); xtaowang@163.com (X.W.)

**Keywords:** ochratoxin A, detoxification, biological method, degradation

## Abstract

Ochratoxins are the secondary metabolites of *Penicillium* and *Aspergillus*, among which ochratoxin A (OTA) is the most toxic molecule. OTA is widely found in food and agricultural products. Due to its severe nephrotoxicity, immunotoxicity, neurotoxicity, and teratogenic mutagenesis, it is essential to develop effective, economical, and environmentally friendly methods for OTA decontamination and detoxification. This review mainly summarizes the application of technology in OTA prevention, removal, and detoxification from physical, chemical, and biological aspects, depending on the properties of OTA, and describes the advantages and disadvantages of each method from an objective perspective. Overall, biological methods have the greatest potential to degrade OTA. This review provides some ideas for searching for new strains and degrading enzymes.

## 1. Introduction

Ochratoxins, as one of the five major mycotoxins (aflatoxins, ochratoxins, fumonisins, zearalenone, and patulin), are secondary metabolites of the *Aspergillus* and *Penicillium* genera. The serious toxicity and carcinogenic effects of ochratoxins make them harmful to the food industry and agriculture [1]. The ochratoxin family has more than 20 subtypes, including ochratoxin A (OTA), ochratoxin B (OTB), ochratoxin C (OTC), non-amide ochratoxin α (OTα) of OTA and OTC, non-amide ochratoxin β (OTβ) of OTB, and hydroxylated OTA and hydroxylated OTB (4R-OH OTA, 4S-OH OTA, 4-OH OTB, 10-OH OTA) [2]. These ochratoxins only differ in structure by a few functional groups, but there is a significant difference in their toxicity, with OTA being the most toxic. OTA is formed by the linkage of L-phenylalanine and isocoumarin via an amide bond. The chlorine atom in the dihydroisocoumarin ring is the main source of OTA’s toxicity, while the chlorine atom in OTB is replaced by a hydrogen atom, meaning its toxicity is only one-tenth that of OTA. Similarly, OTC, Otα, and OTβ do not show significant toxicity [3]. As the most common ochratoxin type, OTA contamination exists widely during the growth, storage, and transportation of different foodstuffs and feeds (cereals, spices, dried fruits, wine, beer, milk, coffee, etc.), posing a major threat to global public health [4].

Upon oral entry into humans or animals, OTA is rapidly absorbed by the gastrointestinal tract into the blood and then metabolized through enzymatic hydrolysis and cytochrome P450 induction. OTA is hydrolyzed to OTα by gut microbes through enzymatic hydrolysis, and the cytochrome P450 system present in the liver mainly hydroxylates OTA to various forms of OTA-OH. Different metabolites are subsequently excreted through urine and feces. However, due to the strong affinity between OTA and serum albumin, which is related to the species, it takes a long time for OTA-OH to be excreted from the body, and the clearance rate of OTA-OH is higher than that of OTα [5].

OTA is associated with a variety of diseases in animals and humans. It has been reported that OTA is nephrotoxic, hepatotoxic, immunotoxic, neurotoxic, mutagenic, teratogenic, carcinogenic, etc. [6] Therefore, it was classified as a Class 2B carcinogen by the World Health Organization’s International Agency for Research on Cancer (IARC) in 2016. OTA is delivered through the food chain and can be found in human and animal tissues or organs. The kidney is the main target organ for OTA toxicity, which can lead to the kidney disease Balkan endemic nephropathy (BEN), urinary tract tumors (UTTs), chronic interstitial nephropathy (CIN), and other serious health problems, but the mechanism of its nephrotoxicity remains unclear. It may be related to the inhibition of protein synthesis, the induction of oxidative stress, and the inhibition of mitochondrial respiration [3,7,8]. Although the process of OTA-induced reactive oxygen species (ROS) production is not well understood, it is generally believed that ROS can play toxic roles by including lipid peroxidation, reducing antioxidant enzyme activity, and inducing DNA counts. Roberto Ciarcia’s team [9,10] discovered in rats that the underlying mechanism of OTA-induced kidney disease is oxidative stress, and in pigs, OTA causes kidney cancer by inducing lipid peroxidation in the proximal tubules. Further studies showed that mitochondrial recombinant manganese-containing superoxide dismutase (rMnSOD) could prevent OTA-induced hypertension and restore lipid peroxidation levels and histological damage in the kidneys of rats [11]. The antioxidant molecules δ-tocotrienol, red orange extract, and lemon extract can reduce or prevent OTA-induced nephrotoxicity through the direct scavenging of reactive oxygen species, which provides a basis for the use of antioxidants to counter OTA’s toxicity [12,13]. The metabolism of OTA is mainly concentrated in the liver. Liver damage caused by OTA recycling in the liver and intestine in vivo has also been extensively reported, which may be due to the oxidative stress of DNA induced by OTA damaging hepatocytes; at the same time, it leads to hepatic steatosis through the PPAR-γ-CD36 axis and affects hepatic lipid metabolism [14,15]. Similarly, antioxidant molecules such as curcumin can significantly reduce OTA-induced oxidative damage and lipid metabolic disorders by enhancing liver catalase activity and remodeling the intestinal microenvironment and metabolism in ducks [16]. The immunotoxicity of OTA is mainly reflected in immunosuppression, which can impair the induction of interferon and reduce the activity of NK cells. Studies have shown that female BALB/C mice exposed to OTA had a significantly reduced proportion of mature CD4+ and CD8+ T cells, inhibiting antibody production. At the same time, it was found that short-term postpartum exposure to OTA could stimulate the immune response [17]. However, it is not clear whether the stimulation is general or specific.

OTA contamination can occur in a wide range of grains and Chinese herbal plants, which poses a potential threat to human health and the economy. Therefore, many countries and regions have implemented limit control measures for OTA. The European Union has set a series of regulations for ordinary materials: a limit of 5.0 μg/kg for unprocessed cereals like some plant seeds; a limit of 0.50 μg/kg for dietary foods for special medical purposes intended for infants and young children; and a limit of 10.0 μg/kg for dried herbs [18]. This review comprehensively sums up the applications of different methods in OTA prevention, removal, and detoxification.

It is important to remove OTA by controlling the generation of mycotoxins at the source. Before harvest, it is essential to implement good agricultural practices (GAPs), like planting antifungal varieties, using bacteriostasis or biological control appropriately, and harvesting in time to reduce the occurrence of fungal infection and control the growth of fungi. When harvesting, it is necessary to ensure the cleanliness of the harvesting equipment and screen and classify products in terms of quality, while damaged or over-mature products should be discarded in a timely manner [19]. In the postharvest phase, the humidity, temperature, and gas composition during grain or feed storage are key determinants of OTA production. In general, the higher the temperature, the higher the humidity, and the more active the fungal growth. *Aspergillus ochraceus*, *Aspergillus carbonarius,* and other related species of *Aspergillus* are the main forces producing OTA during the pre- and postharvest periods of grains, herbal medicine, and feed. The optimal temperature for fungal growth varies, and the minimum humidity and temperature for OTA production are 0.8 a_w_ and 3 °C. Therefore, harvested grain can be treated with hot-air dryers to reduce the water content to about 0.8 a_w_, and after cooling, it can be stored at a relatively low temperature to inhibit the growth of fungi [20,21]. Controlled atmosphere storage (CAS) is often used in the storage of Chinese medicinal materials and grains. By comprehensively regulating the gas composition and temperature and humidity parameters in the storage environment, the reproduction of microorganisms and the respiration of Chinese medicinal materials can be inhibited, and the production of mycotoxins can be reduced at the source. Studies have reported that in order to effectively prevent the accumulation of OTA in moist grains, it is necessary to control the content of carbon dioxide in the atmosphere at greater than 50% and suppress OTA pollution using a significant germ tube length [22]. In order to minimize OTA pollution during the harvesting period, it is necessary to implement effective management strategies throughout the production process and establish feasible standards.

However, it is not realistic to completely suppress the growth of fungi through control alone; further processing of contaminated products is required. According to the type of substance used, decontamination methods can be divided into three categories: physical (radiation, heat, adsorbents), chemical (acid, oxidants, salt), and biological (bacteria, yeast, fungi). This review provides an objective and comprehensive summary of these three commonly used detoxification methods.

## 2. Physical Methods

Physical removal of OTA has the advantages of low costs, environmental friendliness, and simple operation and can effectively remove OTA from contaminated food or feed, but, at the same time, the nutritional matrix or taste of food might be affected due to the non-specific adsorption.

### 2.1. Radiation 

Radiation methods include ultraviolet radiation, gamma radiation, electron beam radiation, and X-ray radiation. Compared with chemical methods, radiation methods can safely and effectively reduce the levels of mycotoxins in food or feed and have been used for food storage on a large scale [23]. Food irradiation was approved by international professional organizations in 1983 as a safe and efficient food processing method [24,25]. The effects of UV light with different wavelengths on the growth, production, and degradation of OTA are significant. Zhang et al. [26] found that UV-B inhibited the growth of *Aspergillus ochraceus* and *Aspergillus carbonarius*, two common OTA-producing strains, at different wavelengths. Shorter wavelengths (blue and purple) significantly inhibited both OTA-producing fungi, while white light only inhibited *Aspergillus ochraceus*. The results showed that the degradation rate of OTA standards could reach 96.50% under UV-B radiation, and the degradation effect of blue light was relatively weak.

The removal of OTA through γ-ray irradiation of artificially contaminated corn showed that the highly reactive radicals produced by irradiation could attack the formation of double bonds of aromatic rings in the OTA molecules [27]. The radiation dose is an important factor affecting the process. At an irradiation dose of 4.5 kGy, OTA production is reduced, while at 6.0 kGy, the growth of *Aspergillus ochraceus* is completely inhibited, and this is the recommended dose to effectively prevent the production of OTA. When the irradiation dose is 20 kGy, the degradation rate of OTA reaches its maximum level of 61.1%. In addition to the γ radiation dose, the effects of different forms of OTA on radiation sensitivity are quite different. By studying the effect of γ-rays on the elimination efficiency of OTA in dry form and different aqueous solutions, it was found that OTA in aqueous solutions was easily degraded by γ-rays, while OTA in dry form was less prone to degradation, indicating that γ rays may not be an ideal method for OTA degradation in grains [28]. UV radiation is often used to detoxify OTA in liquid matrices. However, the detoxification efficiency is unstable because of the influence of the pH value and matrix components, such as amino acids, phenols, ethanol, etc. [29].

It is reported that electron beam irradiation (EBI) offers a higher dose rate capability and can be used to degrade OTA. Peng et al. [30] studied the effect of electron beam irradiation on the degradation efficiency of OTA in an aqueous solution for the first time. The results showed that the degradation process follows a first-order kinetic model, and the degradation efficiency of OTA is dose-dependent, which means the efficiency increases with the increase in irradiation dose and decreases with the increase in OTA concentration. Both the type of solvent and the pH of the solution affect the degradation efficiency. In addition, six products (C_9_H_11_NO_2_, C_13_H_16_O_7_, C_9_H_17_NO_2_, C_22_H_26_ClNO_8_, C_13_H_11_ClO_5_, and C_20_H_19_NO_6_) were identified via LC-MS. It was concluded that ·OH and H· radicals play a crucial role in the degradation of OTA via EBI. Like other radiation technologies, the degradation efficiency of EBI is also affected by the matrix composition and irradiation dose, and there is a risk of reducing nutrients such as vitamins and proteins [31].

Pulsed light (PL) is a sterilization technology introduced in the 20th century that utilizes an instantaneous high intensity of 0.1 to 1 s and broad-spectrum pulsed light energy of 200 to 1100 nm. Wang et al. [32] optimized the response surface methodology, where the number of pulses was 40, the initial concentration of OTA was 50 μg/L, the dilution ratio was 3, and the radiation distance was 2 cm. The degradation rate of OTA in grape juice reached up to 95.29% using PL. Animal experiments showed that PL significantly reduced the toxicity of high-dose OTA in mice, while the LC-MS results showed that PL could remove OTA by degrading it to non-toxic OTα and L-phenylalanine.

Radiation can not only directly inhibit the production of OTA but can also be used as a mutagenic agent to induce mutant strains that degrade OTA. Zou et al. [33] used ultraviolet radiation to screen new strains with an enhanced ability to degrade OTA. The results showed that a new fungus mutant, FS-UV-21, was obtained from *Aspergillus niger*. The virus-free efficiency of the strain in degrading OTA reached 89.4% under certain conditions, significantly reducing the cytotoxicity of OTA. However, the application of radiation methods is limited, and the detoxification effect of OTA on food matrices is not optimistic [28].

### 2.2. Heat

Since the 1990s, heat treatment has been considered a traditional method for OTA removal [34]. The melting point of OTA is 169 °C [35]. According to previous work, although it has thermal stability when the temperature reaches 180 °C or above, the activity of OTA will decrease [36]. Dahal et al. [37] studied the thermal stability of OTA in aqueous buffer solutions under acidic, neutral, and alkaline conditions. It was found that the thermal stability of OTA in the alkaline condition was poorer than in the neutral and acidic buffer systems. Therefore, in alkaline solutions, the OTA removal rate is higher. The removal rate of OTA under alkaline conditions is 50%, while in acidic or neutral environments, the OTA content remains almost stable after processing. Further research has been conducted on the temperature and heat treatment times. The results indicate that they are critical factors in destroying the structure of OTA. H.J. Lee [38] investigated the effect of temperature and time on OTA removal in oats treated at 120 °C and 180 °C for 30 min and 60 min, respectively. With the increase in temperature and treatment duration, the content of OTA in the oats decreased by only 2–18%. At the same time, the thermal stability of OTA will be affected by the moisture content. The presence of moisture will enhance the decomposition ability of OTA, but even when the temperature reaches as high as 200–250 °C, OTA cannot be completely decomposed in grains [21]. In general, the effect of removing OTA from food at high temperatures is not ideal. This is because heat treatment not only induces a large amount of energy consumption but also results in the pyrolysis of other active components, since conservative food processing temperatures range from just 80 to 121 °C [39].

### 2.3. Adsorption

Physical adsorption is an important way to remove OTA. The mechanism is to combine the adsorption material with OTA to form a compound where the mycotoxin is prevented from being absorbed by the body when it passes through the gastrointestinal tract and is eliminated through fecal excretion. The utility model has the advantages of low costs, a simple method, high efficiency, etc. Physical adsorption is a surface phenomenon in which the forces involved include van der Waals forces and electrostatic interactions between the negative charge of OTA and the positive charge of the adsorbent. The phenolic hydroxyl groups in the molecule interact with the hydrophobic materials through π-π bonds. The factors affecting the adsorption effect include the pore size, surface charge amount, charge distribution, and specific surface area of the adsorbent [40,41]. According to the material classification, adsorbents can be divided into natural material adsorbents, inorganic mineral adsorbents, organic synthetic adsorbents, and synthetic composite adsorbents.

#### 2.3.1. Natural Material Adsorbents

Taking advantage of their low cost, wide applicability, and recyclability, natural materials can be used as adsorbents for OTA removal in liquid matrices. Most of these materials come from plants, especially agricultural by-products such as jujube stones, oyster mushroom powder, dried fruit shells, olive pomace, and other fruit wastes that have been reported in the literature [42,43,44]. Adsorbents adsorb fungal toxins on the surface of the material through electrostatic forces, hydrogen bonds, and van der Waals forces, reducing absorption in the gastrointestinal tract and purifying the liquid matrix. Loffredo et al. [45] systematically compared a variety of low-cost natural materials, specifically ground nuts, coconut fiber, waste coffee grounds, and citrus peel. The adsorption of OTA on the liquid substrate (ethanol/water mixture = 14/86, *v*/*v*) demonstrated the great potential of using plant waste as a biosorbent to reduce OTA in liquid substrates. Subsequently, Fernandes et al. [44] investigated the in vitro adsorption effect of micronized olive pomace and grape stems on three common mycotoxins: aflatoxin B_1_ (AFB_1_), ochratoxin A, and zearalenone (ZEN). The results showed that the two adsorbents exhibited higher OTA adsorption efficiency (>89%) at a lower pH (pH = 2), among which the grape stems exhibited a stronger adsorption effect, as 90% of the mycotoxins at a concentration of 10 mg/mL were absorbed. However, the authors only changed the pH value during their in vitro experiments, while the detoxification process in the gastrointestinal environment is complex. Therefore, Nobre et al. [43] further simulated the in vivo gastrointestinal environment and studied the in vitro gastrointestinal detoxification effect of powdered pleurotus otreatus (PO) on OTA and ZEN. They found that an OTA hydrolase present in PO can hydrolyze and detoxify OTA, while PO removes ZEN through adsorption. In addition, the absorption effect of the toxin was greatly reduced when the feed matrix was present, speculating that certain feed matrices could also adsorb mycotoxins and improve the detoxification ability of PO. In general, natural materials represented by agricultural by-products can be used to remove OTA through adsorption, and minority materials can be used to remove OTA through hydrolysis.

#### 2.3.2. Inorganic Mineral Adsorbents

Inorganic mineral adsorbents exhibit good OTA adsorption performance due to their large specific surface area and ion adsorption capacity and are mainly represented by activated carbon (AC), aluminum silicate, hydrated sodium calcium aluminosilicate (HSCAS), bentonite (BEN), zeolite, diatomaceous earth, sea foam, etc. [46,47,48].

AC is most commonly used for the adsorption of OTA in liquid matrices. The European Commission set a maximum dosage of 1 g/L for its use in brewing and refining [49]. Its adsorption characteristics and efficiency are mainly determined by the size, structure, and surface distribution of the pores. AC has a significant adsorption effect on OTA in wine. Var et al. [47] treated 5 ng/mL OTA-contaminated PBS and wine samples with 1 mg/mL AC, where the adsorption efficiency reached 100% and 87%, respectively. However, due to the non-specific adsorption of AC, in addition to adsorbing toxins, it also reduces the characteristic components in wine, such as anthocyanins and phenolic compounds. Therefore, Cosme et al. [48] further studied the influence of AC on wine quality during the removal process. The anthocyanins in red wine reduce the OTA adsorption efficiency of AC by competing with OTA for AC mesopores. For the best adsorption effect, the AC pore volume distribution should fall in the range of 42.6–55.9 Å or 125.6–137.4 Å. Therefore, a proper pore size distribution is crucial for removing OTA from different wine matrices. Through research on the adsorption of AC, a variety of modified ACs have been developed, such as dietary activated carbon and activated carbon fibers [50,51].

BEN is another common OTA mineral adsorbent, which can be divided into calcium bentonite and sodium bentonite according to the type of interlayer cations [47,52]. In addition, modified BEN, tri-octahedral BEN, etc., have also been developed to reduce the OTA content in wine and poultry feed, having made great breakthroughs [53,54]. Rasheed et al. [55] reported the preparation of an organic-inorganic hybrid BEN (OP BEN) modified with orange peel extract. This material can be used to stably adsorb various mycotoxins, including AFB_1_, fumonisin B_1_, and OTA, over wide pH and temperature ranges. This material’s maximum adsorption capacities for OTA and AFB_1_ are 2.13 mg/g and 3.56 mg/g, respectively, and it can be used as an efficient, green-modified material adsorbent for mycotoxins.

Generally, clay-based mineral adsorbents (such as HSCAS) have negatively charged hydrophilic surfaces; thus, their adsorption capacity for the weakly polar toxin OTA, which is also negatively charged, is obviously insufficient [56]. In addition, most of the current research on mineral materials for OTA adsorption focuses on liquid matrices, while the real digestive environment of the gastrointestinal tract is often more complex and changeable. Therefore, more in-depth research is needed to fill the gaps in this area [57].

#### 2.3.3. Organic Synthetic Adsorbents

Organic synthetic adsorption materials are an important component of OTA physical adsorbents, which have attracted in-depth research due to their diverse types, high repeatability, and good adsorption effects. Currently, materials used as adsorbents have been developed and reported, such as modified silica gel, cellulose polymers, and cross-linked chitosan [40,58,59]. Appell et al. [60] synthesized a cyclodextrin-polyurethane polymer and studied its adsorption of OTA in a liquid matrix. The results showed that this material has the ability to reduce OTA up to 10 μg·L^−1^ in red wine samples to levels below the recommended levels (2 μg L^−1^).

With the integration and development of multiple disciplines, a variety of applied composite materials have been designed and synthesized for the physical adsorption and detoxification of OTA. Nanotechnology has been combined with adsorbent materials to develop various graphene derivatives, organic-inorganic hybrid nanomaterials, metal-organic frameworks, nano-adsorbents, and magnetic nano-composite materials [61,62,63]. Alford et al. [64] developed a new type of clay polymer nanocomposite (CPN) that achieves removal by combining with OTA molecules in fruit juice or wine to form a complex precipitation separation. By changing the structure of the polymer, the OTA adsorption and removal efficiencies of the CPNs were optimized. The experimental results showed that the amount of OTA in grape juice absorbed by the CPNs was three times that of montmorillonite (MMT), the adsorption rate was 2–4 orders of magnitude faster than that of MMT, and the nutrient and volume loss of grape juice was also significantly reduced, further highlighting the potential of composite materials in OTA removal. Muhammad et al. [63] prepared a magnetic carbon nanocomposite using sugar bed waste as the main raw material. This adsorbent can be used as a substitute for activated carbon in the detoxification of OTA animal feed and chicken intestinal mucosa. With the deepening of interdisciplinary research on adsorbent materials, more and more materials with good adsorption effects, simple synthesis processes, and green and efficient properties have been developed for the detoxification and removal of OTA.

### 2.4. Other Methods

Cold plasma is the fourth state of matter. As a new technology applied in 2008, cold plasma can effectively degrade and reduce the production of mycotoxins in food and feed [65]. Casas-Junco et al. [66] treated OTA-contaminated coffee samples with cold plasma at a 30 W input power and 850 V output voltage with a helium flow rate of 1.5 L/min for 6 min. The toxin-producing fungi were completely inhibited by artificial inoculation, and the OTA content in the coffee was reduced by 50% after 30 min of sample treatment. The results of the toxicity evaluation showed a decrease from “Toxic” to “Mildly toxic”. The technology of cold plasma is based on the generation of reactive substances, such as O_2_, O_3_, OH, NO·, and NO_2_, which destroy the structure of OTA and, at the same time, reduce oxidative stress, resulting in the degradation and transformation of OTA [67,68]. Cold plasma has the advantage of being able to quickly and efficiently degrade OTA, but its application is limited, to some extent, by the special equipment required, meaning it is rarely applied in toxin degradation.

With the development of technology, more and more physical methods are being applied to degrade OTA or inhibit the growth of fungi, but several limitations exist: incomplete elimination of OTA, nutrient loss or taste change, equipment limitations, etc.

## 3. Chemical Methods

Chemical methods mainly involve destroying the toxic group or changing the solubility of the toxin, and these methods generally hydrolyze the lactone ring or amide bond of OTA. Various kinds of substances have been used for the detoxification of OTA in food and feed. Inorganic acids are represented by formic acid and hydrochloric acid; organic acids mainly include lactic acid, citric acid, and acetic acid [69]. Basic compounds include sodium-hydroxide and potassium carbonate [70]. Strong oxidizing substances, such as hydrogen peroxide, and other reducing substances and salts are also used to degrade OTA [71,72,73].

### 3.1. Alkalization

Under strong alkaline conditions, the amide bond of OTA can be hydrolyzed to nontoxic OTα and phenylalanine, which is a reversible process. In the early stages of the research (as early as 1981), ammoniation was the most common OTA detoxification method [74]. This method has detoxification effects on various OTA-contaminated crops, such as corn and wheat, and does not lead to the production and accumulation of toxic decomposition products [73]. When wheat is ammoniated under high pressure (60 psi) and at a normal temperature, the concentration of mycotoxins can be reduced by 79% [6]. Amézqueta et al. [75] found that 83% of OTA can be reduced by treating contaminated cocoa shells with a 2% potassium carbonate solution at 90 °C for 10 min. Jalili et al. [76] simultaneously investigated the degradation of OTA in black and white pepper by five alkaline compounds (ammonia, sodium bicarbonate, sodium-hydroxide, potassium hydroxide, and calcium hydroxide). The highest degradation rate achieved by sodium h-droxide reached over 50%, while there was no significant difference in the degradation rates between the different alkaline compounds. However, the application of the ammoniation method in the field of crop and poultry feed detoxification is limited because of the long durations and high costs required to achieve the ideal detoxification effect and the reduction in nutritional value [77].

### 3.2. Oxidization

Apart from ordinary oxidants such as hydrogen peroxide and sodium hypochlorite, ozone, as a strong oxidant, has received more and more attention in recent years because it can degrade OTA without the production and accumulation of harmful substance residues. Qi et al. [78] studied the effect of ozone treatment on the degradation of OTA and ZEN in maize and the quality of maize. The degradation rate of ZEN was 100% at 5 s and 65.4% at 120 s after ozone treatment, and the degradation efficiency increased with the increase in ozone concentration, but there was no significant change in the quality of maize after ozone treatment. The mechanism of mycotoxin inactivation by ozone is related to the types of mycotoxins. It has been reported that ozone can react with functional groups in mycotoxin molecules, resulting in structural modifications [79]. However, the mechanism of OTA degradation by ozone still requires further study. Ozone, as a toxic gas, should be used in a closed environment while strictly controlling the dosage to reduce harm to human health [80].

### 3.3. Other Chemicals

OTA can be converted to OTα and L-phenylalanine via heating under acidic conditions [81]. However, compared with other chemicals, this decomposition effect is weaker, and the degraded products are unstable. The reduction in OTA in black pepper using chloridric acid was only 33.7% [76]. A 2020 report pointed out that organic acid is better than inorganic acid in reducing OTA in grape residues, inferring that polyphenols have an impact on the reduction in OTA through the acid reduction process [69]. 

Salt is also used for the detoxification of OTA. Like sodium hyposulfite, sodium bisulfite has been proven to have an excellent degradation effect on OTA, but sodium chloride has little effect on the reduction in OTA content [72,76].

The accumulation of toxic chemicals will cause damage to humans and the ecological environment, affecting the nutrients in food. Therefore, the use of chemicals, such as fungicides, requires strict compliance with regulations to ensure that pesticide residues in food are below the maximum levels set by the European Commission. In addition, some fungicides can also cause toxin-producing fungi to increase fungal synthesis [82], so the need for new reagents is increasing. Magista et al. [83] studied the effect of active chlorine-electrolyzed oxidizing water (EOW) produced using KCl on the removal of OTA from *Aspergillus carbonarius*-infected grapes. EOW treatment reduced the infection rate by about 87–92% in detached berries, and the OTA concentration decreased by 92% in vitro, showing a better effect than the fungicide Switch. This study shows that EOW can replace fungicides and reduce OTA produced by *Carbonarius*, which plays an important role in grape contamination, thus reducing the side effects of agricultural residues.

## 4. Biological Methods

Biological methods are based on non-toxic bacteria, yeast, fungi, and other natural microorganisms as the main components and work by inhibiting the growth of toxin-producing fungi, binding mycotoxins, and degrading OTA to non-toxic products such as OTα in contaminated matrices. With the increasing awareness of environmental protection and food safety, biological methods have many advantages, including, but not limited to, low costs, obvious effects, minimal side effects, and environmental friendliness, and thus have been widely applied in reducing OTA in food, grain, and feed. According to the mechanism of action, biodegradation methods can be categorized into three types: inhibition of fungi for preventing OTA production, and biosorption and enzymatic degradation in OTA-contaminated matrices for detoxification. It should be emphasized that microbial detoxification could be achieved simultaneously with the physical adsorption, enzymatic biodegradation, and growth inhibition of mycotoxigenic fungal species. The microorganisms used for biological detoxification need to meet the following basic requirements: they must be non-pathogenic, have a low growth demand, be able to effectively degrade OTA, and not generate toxic metabolites [84]. Based on their high degradation efficiency and unique advantages, biological methods have been used as the main means of removing OTA.

### 4.1. Inhibition of OTA Biosynthesis

Microorganisms can inhibit OTA production in a number of ways. Anti-microbial lipopeptides from *Bacillus subtilis*, like surfactin, fengycins, and iturins, play an important role in the wine production process. On the one hand, they can promote the synthesis of esters and acids to promote fermentation while ensuring the nutritional value and taste are unaffected [85]. On the other hand, they inhibit the production of OTA through a variety of mechanisms. Iturins are lipopeptide biosurfactants synthesized and secreted by *Bacillus subtilis*. Iturnin A, one of the iturin homologues, inhibits OTA by altering the cellular structure of *Aspergillus niger*, causing a metabolic imbalance, and inhibiting spore germination [86]. *Lactobacillus* is a kind of bacteria that can inhibit the production and growth of mycotoxins; moreover, it has beneficial effects on human health. Li et al. [87] isolated *Lactobacillus brevis strain* 8–2B, which significantly affects the growth, spore germination, germ tube length, and OTA production of *Aspergillus carbonarius*. This strain inhibits the production of OTA by destroying the cell structure of *Aspergillus carbonarius* and down-regulating the genes related to OTA synthesis. Yeast, as an important microorganism in OTA biocontrol, has several advantages, such as its diverse species, simple nutrient requirements, rapid growth, and the fact it does not produce allergenic compounds or secondary metabolites [88]. It has the ability to reduce the OTA production of related fungi by producing volatile organic compounds (VOCs) and to inhibit hypha growth and spore germination and formation. Farbo et al. [84] revealed that the yeasts *Cyberlindnera jadinii*, *Cyberlindnera friedrichii*, *Cyberlindnera intermedia,* and *Lachancea thermotolerans* inhibit the OTA production of two *Aspergilli species* by releasing VOCs, of which 2-phenylethanol is the major component of yeast VOCs. In addition, VOCs could affect gene expression related to OTA production in fungi. Overall, there are two main mechanisms through which OTA production is reduced by controlling OTA-producing fungi: (1) altering cell metabolism, inhibiting cell proliferation, and destroying the cell structure; and (2) inhibiting the gene expression involved in OTA biosynthesis in fungi.

### 4.2. Microbial Bioadsorption

At present, the microorganisms used in biosorption include yeasts [89,90], bacteria, and fungi [91,92,93]. The main factors affecting the adsorption method are the nature of the microorganisms, including the types of microorganisms, culture conditions [94], and genetic characteristics [95]. 

*Saccharomyces cerevisiae* is a representative of biological OTA removal, and its mechanism is adsorption. β-glucan and mannose proteins are major components of the yeast cell wall; thus, negatively charged mannose proteins bind to OTA through polar and nonpolar interactions. Due to the degree of glycosylation of cell wall changes depending on the species of yeast, the ability of different strains to adsorb OTA varies [96]. There are research reports indicating that inactivated *Lactobacillus brevis*, *Lactobacillus plantarum*, and *Lactobacillus sanfranciscensis* [97] showed a better OTA removal efficiency in the medium than in live cells; furthermore, the mechanism of OTA removal involves OTA being adsorbed to the cell wall of *Lactobacillus* through hydrophobic and acid-base interactions. Some compounds have been found to be useful in reducing OTA-induced toxicity. Based on the premise of using N-acetyl-L-cysteine (NAC) to reduce the toxicity of OTA and AFB_1_ to pig alveolar macrophages, one team studied NAC’s negative effects on OTA toxicity and the synergistic effect of *cryptococcus podzolus Y3* on OTA degradation. It was found that after 24 and 48 h of cultivation with 10 mM NAC, the OTA degradation rates of *cryptococcus podzolus Y3* increased by 100% and 92.6%, respectively, providing an efficient and sustainable strategy for improving the OTA degradation efficiency [98].

### 4.3. Enzymatic Degradation of OTA

As an important component of the OTA biodegradation mechanism, enzymes are mainly derived from the production of specific microbial strains. Firstly, degradative enzymes are screened from strain cells, and then the target protein is obtained using exogenous expression technology. The enzymes produced can be both extracellular enzymes and intracellular enzymes [99]. The most common types of OTA biodegradation enzymes are mainly carboxypeptidase and amidase. They can degrade the OTA-amide bond or lactone ring into the less-toxic OTα or ring-opened OTA to reduce the toxicity, as shown in Figure 1. Furthermore, other common degradation mechanisms are enzymatic hydrolysis of the amide bonds in OTA molecules to L-β-phenylalanine and non-toxic OTα, or hydrolysis of lactone rings to achieve detoxification. Cortes et al. [100] used molecular docking technology to predict the potential application of Ananas comosus bromelain cysteine-protease, bovine trypsin serine-protease, and *Bacillus subtilis* neutral metalloendopeptidase in OTA detoxification. The results of in vitro experiments show that bromelain and trypsin have a certain OTA detoxification ability under acidic conditions, and the metalloendopeptidase is a highly efficient OTA biodetoxifier, which proves that OTα is the final product of the enzymatic reaction.

Microorganisms that have been found to produce amidases that degrade OTA include *Aspergillus niger* [101], *Alcaligenes faecalis* [102], and *Stenotrophomonas acidaminiphila* [103], which produce the less toxic OTα and phenylalanine by hydrolyzing the amide bond in OTA. Liang et al. [104] separated *Brevundimonas naejangsnensis* ML17 strain and obtained four new OTA and OTB degradases, namely BnOTase1, BnOTase2, BnOTase3, and BnOTase4. These enzymes degrade OTA and OTB to OTα and OTβ through hydrolyzed molecular lactide, and the degradation rate is as high as 100%. Luo et al. [103] isolated and identified the OTA degradase ADH3 from *Stenotrophomonas acidaminiphila*, a strong temperature-adapted amide hydrolase (0–70 °C) that can completely convert 50 mg/L OTA into OTα within 90 s (1.2 mg/mL). Its catalytic efficiency is 56.7–35,000 times higher than that of rAfOTase, rOTase, and commercial carboxypeptidase A (CPA). ADH3 is the most active enzyme for OTA degradation reported thus far.

Another mechanism of OTA-degrading enzymes involves opening the lactone ring of the OTA molecule through hydrolysis to generate the final degradation product OTA with the lactone ring open (OP-OTA), such as CPA [105], ochratoxinase (OTase) [106], carboxypeptidase PJ 1540, and other hydrolytic enzymes [92,107]. Xu et al. [105] reported that *Bacillus subtilis CW14* has the ability to detoxify OTA. By analyzing the signal peptide of the genomic group, several extracellular enzymes, such as carboxypeptidase, hydrolase, and amylase, were predicted to be related to OTA detoxification. The results also showed that the virus-free rate of *Escherichia coli* expressing the DACA and DACB proteins during OTA degradation reached 71.3%, revealing the huge potential in agricultural products and feed production. In order to solve the problems with the poor stability and non-recovery of the above-mentioned hydrolases, Ma et al. [108] adopted immobilization technology. Immobilization of polyvinylpyrrolidone CPA in zeolitic imidazolate framework materials improves the stability and degradability of CPA. The immobilized enzyme can be reused more than 10 times, and the OTA degradation rate is 30.69% higher than that of free CPA, indicating that immobilization is an effective way to improve the degradation ability of the enzyme.

The use of enzyme biodegradation methods has the advantages of repeatability, a simple treatment process, and the absence of secondary pollution [109], but there is a risk of forming a toxic intermediate [110]. Freire et al. [111] proposed that the metabolism of OTA in yeast strains includes four types: plochide, hydrolysis, hydroxygenization, and combination. The main products include OTα, OTβ, OTα methyl ester, OTB methyl ester, ethylamide OTA, OTC, hydroxy-OTA, hydroxy-OTA methyl ester, and OTA cellobiose ester. In addition to the common low-toxicity products, the author also believes that some OTA-opening-loop products are more toxic than OTA. The results of the study indicate that the degradation products and specific toxicity should be clarified when using yeast as an OTA detoxification agent.

As the mainstream method for detoxifying OTA, biological methods still have some shortcomings, such as the long processing cycle, the impact of the growth of metabolic products of microorganisms on the taste of the product, and the limited application scope [112].

## 5. Conclusions

OTA contamination in food and crops has attracted extensive attention. At present, the methods commonly used in OTA removal or detoxification have defects to some extent. The main shortcomings of physical absorption technology are that the adsorption efficiency is lower as a whole, the scope of application is limited, and the adsorption of OTA has no specificity. Therefore, new degradation materials can be designed and developed based on the characteristic groups in the OTA structure, such as hydroxyl, carboxyl, chloride, and lactone groups. By integrating nanoscience, the degradation rate of OTA can be improved, and the problem with adsorbent residues can also be solved. As for chemical methods, the main problem is toxicity. Solving the toxicity and environmental problems caused by chemical residues is an urgent task. Due to its advantages, such as its low toxicity, high efficiency, and environmental friendliness, the biodegradation of OTA has become increasingly popular. In addition to the two degradation mechanisms summarized above, some different, unclear degradation processes have been found. Clarifying unknown mechanisms for screening new strains for OTA degradation is also an important task. Moreover, it is essential to shorten the detoxification cycle while operating on a large industrial scale. In the future, biodegradation combined with genetic engineering, protein engineering, and site-directed mutagenesis, among other technologies, will have broad prospects.

## Figures and Tables

**Figure 1 toxins-15-00565-f001:**
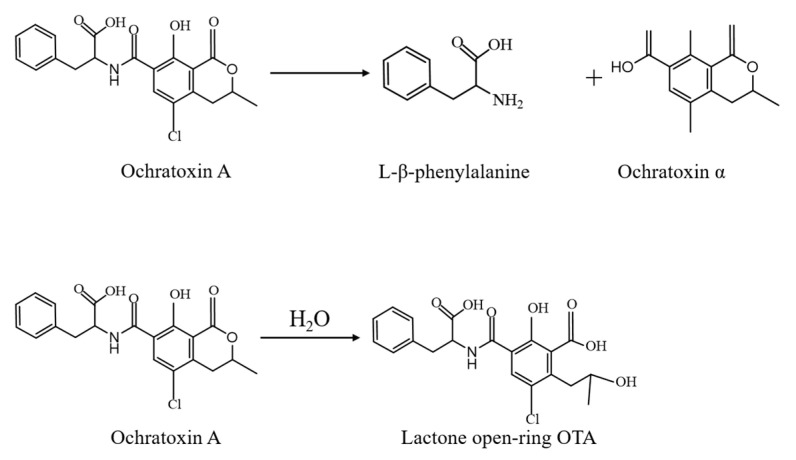
The biodegradation mechanism of OTA.

## Data Availability

No new data were created.

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
