# Peer review of "Ochratoxin A: Overview of Prevention, Removal, and Detoxification Methods"

_toxins, 2023, doi:10.3390/toxins15090565_

Round 1
Reviewer 1 Report
This manuscript is a review article focusing on the prevention, removal and detoxification methods for Ochratoxin A, one of the five legislated mycotoxins, so that there is no new data generated per se. The topic is relatively focused so the review is short at 13 pages, including 92 references. Of those references some 56 or so are from the last five years so that the manuscript deals with relatively up-to-date literature.
The organization is good, dealing in turn with physical, chemical and biological methods to detoxify or remove Ochratoxin A from contaminated food or feed, but the titles on pages 2 and 6 should be pluralized (e.g. Chemical methods) because the authors are discussing more than one method within each classification.
There are numerous grammatical errors, too many to outline as well as poor sentence structure and inappropriate terms. Please remove all references to “besides” and “and so on”. When referring to quoted references in the text quote as “first author et al.” rather than a single author or group and don’t use first names (see page 3, lines 104, 137, 142 and 149; page 4, lines 167, 171 and 199; page 7, lines 325 and 337; page 8, line 386; page 9, line 405 and elsewhere).
The authors would benefit from having someone proficient in English review the whole paper and correct all grammatical, tense and sentence structure issues.
Specific comments;
Page 1: abstract, line 11: This is not a sentence – lacks a verb, line 12: This sentence is not constructed properly – do the authors mean that “Biological methods hold he greatest potential for OTA degradation”?, line 17: why is “mycotoxin” here? – they are all mycotoxins, lines 23-24: poor grammar, line 27: sentence structure, line 39: “accumulation”.
Page 2: line 48: “how to precisely detect and remove OTA”, lines 74-76: grammar
Page 3: line 108-109: sentence structure, line 115: ”pyrolysis”, line 122: omit “Johannes Diderik” – not relevant.
Page 4: lines 155-157: the sentence is confusing, lines 201-202: 10 ug/L is not lower than 2 ug/L?
Page 5: line 237: “toxic to mildly toxic”
Page 6: lines 270-271: Do not start a sentence with “because” – maybe this can be joined with the previous sentence.
Page 7: lines 308, 315-316: grammar, lines 318-340: this whole paragraph should be rewritten. Iterins are compounds produced by B. subtilis, according to reference 66.
Page 8: line 379: What is this line doing here?
Page 9: lines 404-407: this sentence is confusing, line 432: “research hotspot” is not a technical term, maybe “more popular”?, lines 432- 438: these two sentences need to be corrected for grammar – the final sentences in a conclusion should be clear and concise.
There are numerous grammatical errors, too many to outline as well as poor sentence structure and inappropriate terms. Please remove all references to “besides” and “and so on”. The authors would benefit from having someone proficient in English review the whole paper and correct all grammatical, tense and sentence structure issues. See specific comments above.
Reviewer 2 Report
The review provides a good overview about the recent technologies for OTA detoxification methods, although much is known about it. However, it needs to be expanded, especially by focusing on toxicity to humans and animals. I therefore propose to update this review. For example, adding information present in these scientific publications: DOI: 10.1002/jcp.26753; DOI: 10.1016/j.psj.2019.10.041; DOI: 10.1002/jcb.25425; DOI: 10.3390/toxins14020067; DOI: 10.1080/15287390500195570.
Moreover, there are some types error that need to be corrected and some expressions that need to be remodelled.
Reviewer 3 Report
This work entitled “Ochratoxin A: overview of prevention, removal and detoxification methods” is focused on OTA detoxification by physical, chemical and biological strategies reviewing recent studies. In my opinion, the study is a difficult to read in some paragraphs, but in general technically sounds. I would suggest that at least some general information should be provided on the date since some cited decontamination techniques were investigated or used, to highlight the new techniques from the old ones.
- The authors should specify in the text if decontamination methods were applied in microorganisms for preventing OTA production or in OTA contaminated matrices for detoxification.
- The title of this study should be changed because “prevention” methods are omitted through the manuscript, even in a general way. This information could be added after the introduction section: control strategies (strategies to reduce OTA levels in cereal, grains or feed during storage), and good agricultural, hygienic and manufacturing practices to avoid mould infestation and thus mycotoxins production during pre- and postharvest periods.
- In my opinion, Abstract should be rewritten and refined. There are some inaccurate sentences, for instance: “This review introduces the properties of OTA…”; “Objectively describes the advantages and disadvantages of each method”.
- In the course of the text, it is necessary to use italic letters in microorganism names identification.
- As OTA is highly resistant to conventional treatments such as thermal processes or fermentation, it is important to remark critically through the text, the best techniques to removal OTA particularly in food matrices, in each of the techniques studied.
- Different previous studies showed not significant effects in detoxification experiments, this manuscript only examined positive results. This fact should be discussed in the text.
The following suggestions should be considered by the authors to improve this contribution:
1. Line 17: Delete “mycotoxin” in this sentence.
2. Line 17: Change “is a secondary metabolite of Aspergillus and Penicillium fungi” with “…Aspergillus and Penicillium genera”
3. Lines 29-30: Complete this sentence as “…during the growth, storage and production of different foodstuffs and feed: cereals, spices, dried fruits, wine, beer, milk, coffee, etc., …”
4. Line 42: Complete this sentence as “OTA contamination can occur on a wide range of grains and Chinese herb plants, which…”
5. Lines 44-47: Please, revise data of maximum level and insert this actualized reference: “Commission regulation (EC) No 1370/2022 of 5 August 2022 amending Regulation (EC) No 1881/2006 as regards maximum levels of ochratoxin A in certain foodstuffs. Official Journal of the European Union, L206/11-206/14”.
6. Line 52: Change “1. Physical method” with “1. Physical methods”. In the same way it should be changed in the following points 2 and 3 in the text.
7. Lines 56-57: Please, delete this sentence “The specific physical removal methods like radiation, heat treatment, adsorption, cold plasma, and so on”.
8. Line 65: “Zhang et al. found that UV-B…” please, change citation in text with “Zhang et al [15] found that UV-B…” and correct this throughout the text.
9. Why the authors not include “pulsed light (PL) method” at radiation subsection?
10. Line 100: Please, change “matrix” with “matrices”.
11. Line 103: Please, change “169ºC” with “169 ºC” and correct through the text.
12. Line 115: Please, change “pyrolysisi” with “pyrolysis”.
13. Lines 122-124: Please, complete this sentence as “…which the forces involved including van der Waals forces and electrostatic interaction between the negative charge from OTA and the positive charge of the adsorbent. The phenolic hydroxyl groups…”.
14. Lines 142,147,149 and 150: Please, insert “in vitro” and in vivo” in italic letters.
15. Line 155: Please, if possible, insert another word instead of “sideline”, its meaning is not clear, in this context.
16. Line 173: Change “activated carbon” with “AC”.
17. Lines 200-202: In my opinion, this sentence it is confusing: “The results showed that this material can significantly reduce the content of OTA in red wine less than 10 µg/L, which is lower than recommended levels (2 µg/L)[43]”.
18. Line 236: Please, change “minutes” with “min” and correct through the text.
19. Line 250: Please, change “otα” with “OTα”.
20. Line 259: Please, rewrite this sentence: “Over the past few decades, various kinds of chemical substances have been discovered for the purification of mycotoxins”. Only as a suggestion: “Over the past few decades, various kinds of chemical substances have been used for the detoxification of food products and feedstuffs”.
21. Lines 304-306: complete this sentence as “…by inhibiting the growth of toxin-producing fungi, and binding mycotoxins, or degrading OTA to non-toxic products as OTα in the contaminated matrices”
22. Line 317: Please, change this caption with “3.1 Inhibition of OTA biosynthesis”.
23. Lines 318-322: This paragraph should be rewritten because it possibly gives rise to an error in the interpretation. Iturin is a lipopeptide biosurfactant synthesized and secreted by Bacillus subtilis. Iturin has an antifungal activity against to filamentous fungi without causing toxic effects (Gao et al. Frontiers in Bioengineering and Biotechnology 10.974460, 2022).
24. Line 341: Please, replace “3.2. Microbial adsorption degradation method” with “3.2 Microbial bioadsorption”.
25. Lines 362-370: This paragraph should be translated to point 3.3.
26. Line 420: It is important to remember that microbial detoxification could be produced simultaneously by physical adsorption, enzymatic biodegradation or inhibiting the growth of mycotoxigenic fungi species. This fact could be cited in the text, at this point or at the beginning of the subsection (3).
27. Line 420: The authors should explain if OTA biodegradation by antagonistic microorganisms was more effective using inactivated cells or live cells.
28. Line 431: In view of the studies shown [91, 92], the claim of “low toxicity” is not substantiated.
29. References section is not formatted according to journal instructions. This section must be rearranged.
m
Round 2
Reviewer 1 Report
The authors have dramatically improved this review article. They have reorganized the review somewhat, adding to the introduction pages and more references before dealing with the physical, chemical and biological methods of prevention/removal/detoxification of OTA. The authors have benefitted from having the revised manuscript edited by a professional editing service of MDPI and virtually all the grammatical errors have been corrected.
However, the authors must pay careful attention to the references. The majority of the 113 references do not have the name of the journal. For those references that do, the journal name is in italics. Many ( see references 1, 2, 4, 5, 11, 22, 32 and more) have the authors, title, year, volume and page numbers but without the journal name, they cannot be looked up. Still others (e.g. references 12, 13, 40 and 76) have a series of initials. Is J. o. f. p. a journal name? Please use the conventional short forms for journal names. These need to be corrected before publication – there is no point of a review unless the reader can look up the papers cited.
Specific comments:
Page 4, line 187: should read “H.J. Lee (39)”
Page 10, line 503: should read “Freire et al. (110)”
Much improved thanks to the use of a professional editing service.
Reviewer 2 Report
The manuscript has been successfully improved and can be considered for publication
Author Response
Thank you for your recognition of our manuscript.
Reviewer 3 Report
The revised manuscript has been improved, but there are some typographical errors that should be corrected. References section should be revised according journal instructions.
JournalArticles:
1. Author 1, A.B.; Author 2, C.D. Title of the article. Abbreviated Journal Name Year, Volume, page range.
- Page 1, line 42: please correct this sentence as “Therefore, it was classified as a class 2B carcinogen by…”
- Page 5, line 249: “Erginkaya et al. [48] treated…” should be changed by “Var et al. [48] treated…”.
- Page 6, line 297: “Khisroon et al. [64] prepared…” should be changed by “Muhammad et al. [64] prepared…”.
- Page 7, line 37: “and normal temperature, …”, please define what it is a “normal” temperature?
- Page 9, line 462: “…and Bacillus subtilis…” please use italic letters in the microorganisms. The same at page 10, line 492.
- Page 10, line 503: “Luisa Freire [112]…” please change with “Freire et al. [112]…”.
u
